# Divergent effects of central melanocortin signalling on fat and sucrose preference in humans

Agatha A. van der Klaauw[1], Julia M. Keogh[1], Elana Henning[1], Cheryl Stephenson[1], Sarah Kelway[1], Victoria M. Trowse[1], Naresh Subramanian[1,2], Stephen O'Rahilly[1], Paul C. Fletcher[1,2] & I. Sadaf Farooqi[1]

Melanocortin-4-receptor (MC4R)-expressing neurons modulate food intake and preference in rodents but their role in human food preference is unknown. Here we show that compared with lean and weight-matched controls, MC4R deficient individuals exhibited a markedly increased preference for high fat, but a significantly reduced preference for high sucrose food. These effects mirror those in *Mc4r* null rodents and provide evidence for a central molecular circuit influencing human macronutrient preference.

[1] Clinical Biochemistry, University of Cambridge Metabolic Research Laboratories and NIHR Cambridge Biomedical Research Centre, Wellcome Trust-MRC Institute of Metabolic Science, Box 289, Addenbrooke's Hospital, Hills Road, Cambridge CB2 0QQ, UK. [2] Department of Psychiatry, University of Cambridge and the Cambridgeshire and Peterborough NHS Foundation Trust Cambridge, Cambridge CB2 0SZ, UK. Correspondence and requests for materials should be addressed to I.S.F. (email: isf20@cam.ac.uk).

Neural circuits involving the hypothalamus, brainstem and mesolimbic system play a pivotal role in the regulation of eating behaviour in response to changes in nutritional state and meal consumption[1,2]. Primary leptin-responsive neurons in the arcuate nucleus of the hypothalamus expressing the anorectic peptide pro-opiomelanocortin (POMC) or the orexigen agouti-related peptide (AgRP) project to the paraventricular nucleus of the hypothalamus where they synapse with neurons expressing the melanocortin 4 receptor (MC4R) to regulate food intake[3,4]. Targeted disruption of melanocortin signalling in *Pomc*, *Agouti* and *Mc4r* null mice leads to an increase in food consumption and more specifically to increased preference for a high fat diet in food choice experiments[5–7]. Studies in $Mc4r^{-/-}$ mice and rodents harbouring a point mutation in *Mc4r*, suggest that disruption of melanocortin signalling paradoxically reduces consumption of high sucrose solutions and food[6,8], an effect that is not seen with the non-nutritive sweetener sucralose, and is not explained by conditioned taste aversion[6]. Interestingly, live optical recording of AgRP and POMC neurons in awake mice has revealed direct activation/inhibition of these neurons by a highly palatable diet offered in the energy-deficient state[9]. While there is considerable data regarding the contribution of central melanocortin signalling to food preference in rodent models, the relevance of these findings to food preference in humans has not been explored. Highly penetrant loss of function *MC4R* variants in humans are rare (1–5% of people with severe obesity), but as the phenotype of human MC4R deficiency closely parallels that seen in mice lacking Mc4r[10], studies in genetically characterized individuals allow us to directly test whether disruption of melanocortin signalling alters food preference in humans. We tested whether MC4R deficient individuals have an altered preference for foods of varying fat and sucrose content. We found that compared with obese and lean controls, individuals with MC4R mutations show an increased preference for high fat food but a decreased preference for sucrose containing food.

## Results

**Fat preference test**. We tested whether MC4R deficient individuals have an altered preference for foods of varying fat and sucrose content, compared with obese and lean controls. We developed a three choice, *ad libitum* meal paradigm in which fat content was covertly manipulated to provide 20 (low), 40 (medium) and 60% (high) of the total caloric content (Table 1; Methods), critically without altering appearance, texture or taste. Twenty lean controls (BMI $23.1 \pm 0.3 \mathrm{\,kg\,m}^{-2}$), 20 obese controls (BMI $34.1 \pm 1.1 \mathrm{\,kg\,m}^{-2}$) and 14 adults with heterozygous loss of function *MC4R* variants (BMI $38.8 \pm 2.2 \mathrm{\,kg\,m}^{-2}$; Table 2) participated; BMI did not differ significantly between the obese and MC4R groups. Results were analysed using either ANOVA analysis (with interaction terms for study group and study meal)

with Tukey's HSD post-hoc comparisons for the fat preference study. Significance was set at $P = 0.05$.

Participants were initially provided with 'tasters' (15 g) of the 3 meals, and liking/disliking ratings obtained using visual analogue scales (VAS) before and after meal consumption. Liking ratings for the low/medium/high fat meals were similar before and after meal consumption for all three groups (Fig. 1a,b). Although liking scores for the high-fat meal were comparable to those for the low and medium fat meals, MC4R deficient individuals consumed 95% more of the high fat meal than lean, and 65% more than obese controls (Fig. 1c,d; $P = 0.0222$). There was no difference in total intake collapsed across fat level (mean ± s.e.m.; g) between lean controls (488.5 ± 43.1), obese controls (551.0 ± 48.8) and MC4R deficient individuals (561.5 ± 60.6); $P = 0.53$ for the group comparison.

**Sucrose preference test**. In the second study, we investigated sucrose preference in a further 20 lean (BMI $22.5 \pm 0.4 \mathrm{\,kg\,m}^{-2}$) and 20 obese controls (BMI $35.9 \pm 1.2 \mathrm{\,kg\,m}^{-2}$) and 10 adults with heterozygous loss of function *MC4R* variants (Table 2, BMI $41.3 \pm 3.2 \mathrm{\,kg\,m}^{-2}$). The test dessert provided 8 (low), 26 (medium) and 54% (high) sucrose; to avoid interference from fat preference in this paradigm, the energy content from fat was clamped at ~30% (Table 3).

We found that lean and obese volunteers liked the high sucrose meal more than the low or medium sucrose meals in keeping with previous studies of sucrose preference in volunteers[11] (Fig. 2a,b). In MC4R deficient individuals, although liking ratings for low and medium sucrose meals were comparable with the lean and obese groups, liking ratings for the high sucrose meal were significantly reduced (Fig. 2a,b; $P = 0.0252$). Individuals with *MC4R* variants also consumed significantly less of all three sucrose meals compared with lean and obese controls (Fig. 2c,d; $P = 0.0064$). In summary, our findings in MC4R deficient individuals indicate that central melanocortin circuits play a key role in modulating fat and sucrose preference in humans.

**Sensory discrimination test**. It is well recognized that oral perception can affect food preference[11]. To test whether there was a perceptible sensory difference between the low (20%; A) and high (60%; B) fat test meals, we employed a sensory discrimination test (Triangle Test). We found that 41 out of the 78 panellists we tested, correctly identified the odd sample in the test. According to international standards for this test

### Table 1 | Macronutrient composition of meals used in the fat preference test.

| Meal | Fat (%) | Protein (%) | Carbohydrates (%) | Caloric density (kcal g$^{-1}$) |
|---|---|---|---|---|
| Low fat | 20 | 25 | 56 | 1.1 |
| Medium fat | 40 | 19 | 42 | 1.5 |
| High fat | 60 | 13 | 28 | 2.0 |

Percentage of energy derived from each macronutrient is shown. Kcal = Calories.

### Table 2 | *MC4R* variants carried by individuals taking part in these studies.

| Study | Variant | Number of subjects |
|---|---|---|
| Fat preference | G252S | 2 |
| Fat preference | Y80X | 1 |
| Fat preference | R236C | 1 |
| Fat preference | I125K | 4 |
| Fat preference | C271Y | 3 |
| Fat preference | Y35X; D37V | 2 |
| Fat preference | I137T | 1 |
| Sucrose preference | I125K | 2 |
| Sucrose preference | E61K | 2 |
| Sucrose preference | C271Y | 2 |
| Sucrose preference | A144S | 1 |
| Sucrose preference | Y35X; D37V | 2 |
| Sucrose preference | F280AfsX12 | 1 |

Data on these mutations has been published previously[20,29,30].

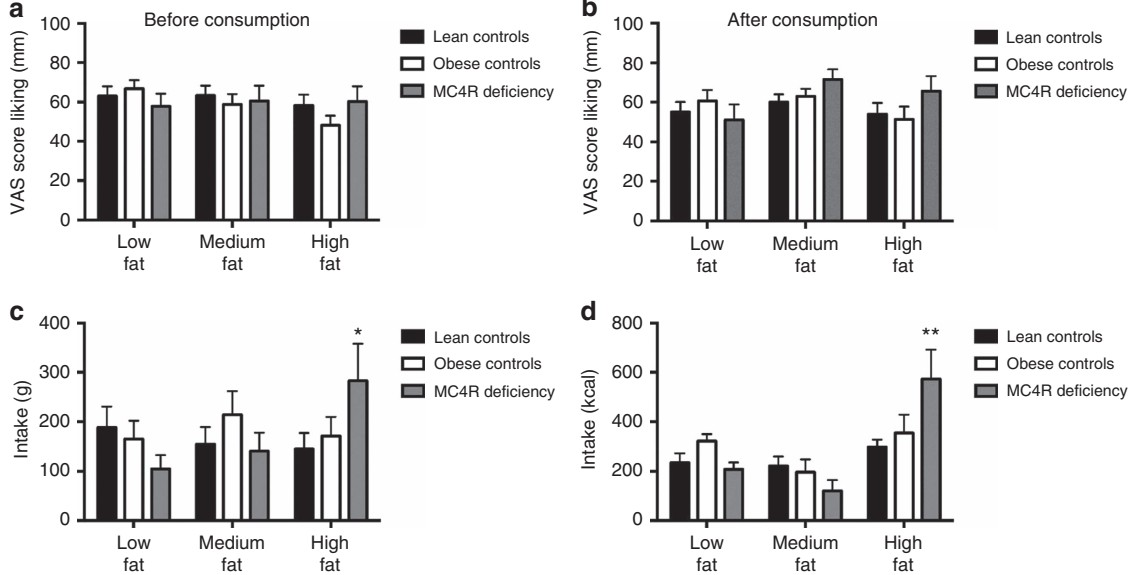

**Figure 1 | Liking ratings and food intake in the fat preference test.** Liking of low, medium and high fat meals before (**a**) and after (**b**) an *ad libitum* test meal in lean controls (*n* = 20), obese controls (*n* = 20) and individuals with MC4R deficiency (*n* = 14). No differences were found between the groups or between meals for liking (VAS = visual analogue scales). (**c**). Total intake (g and kcal; **c,d** respectively) in lean controls, obese controls and individuals with MC4R deficiency. Means ± s.e.m. (error bars) are shown. Results were analysed using ANOVA analysis (with interaction terms for study group and study meal) with Tukey's HSD post-hoc comparisons. *$P$ = 0.0222 and **$P$ = 0.023 for the interaction of group by meal.

| Meal | Sucrose (%) | Protein (%) | Carbohydrates (%) | Fat (%) | Caloric density (kcal g⁻¹) |
|---|---|---|---|---|---|
| | | | | | |

**Table 3 | Macronutrient composition of meals used in the sucrose preference test.**

| Meal | Sucrose (%) | Protein (%) | Carbohydrates (%) | Fat (%) | Caloric density (kcal g$^{-1}$) |
|---|---|---|---|---|---|
| Low sucrose | 8 | 38 | 26 | 36 | 1.0 |
| Medium sucrose | 26 | 25 | 40 | 36 | 1.3 |
| High sucrose | 54 | 11 | 60 | 29 | 2.1 |

Percentage of energy derived from each macronutrient is shown. Kcal = Calories.

(www.astm.org/Standards/E1885.htm), in a study with 78 panellists ($\alpha$ = 0.001) at least 40 correct responses are required to reject the null hypothesis (that is: that there is no difference between the meals). Since we observed 41 correct responses, we cannot conclude that the two meal types are the same, although these results are suggestive. As we were not able to perform this test in a comparable number of people with MC4R deficiency due to the rarity of this disorder, we cannot formally exclude the possibility that attenuated perception of fat may have contributed to the effect that we observed. However, it is noteworthy that on debriefing, participants did not report differences between the foods and were unaware that we had manipulated fat content.

## Discussion

Although common genetic variants in the fatty acid translocase CD36 have been associated with liking of high-fat foods in African Americans[12] and common obesity-associated variants have been associated with diary reports of food choices in a number of studies[13,14], this is to our knowledge one of the first experimental studies to show a direct association between

macronutrient preference (other than alcohol) and a specific genetic/molecular mechanism in humans. Additional studies using a forced choice paradigm which makes a direct comparison between fat and sucrose will be needed to test whether disruption of melanocortin signalling primarily alters sucrose or fat sensing or the reward from these stimuli.

The potential neural mechanisms involved in fat and sucrose preference have been explored in rodent studies. Fat preference may in part be mediated by MC4R-expressing neurons in the amygdala where injection of AgRP acutely increases[15], whereas the agonist Melanotan-II decreases[16], fat consumption in rats. The central nucleus of the amygdala receives and provides projections to the parabrachial nucleus, a region which responds to gustatory and visceral signals, is involved in sucrose preference and is connected to excitatory MC4R-expressing hypothalamic neurons[17–19]. Previous studies have shown that pre-prandial gut hormone levels (total ghrelin, PYY and GLP-1), as well as plasma insulin and glucose are comparable in MC4R deficiency and weight-matched controls[20–22]. Although postprandial total ghrelin suppression was attenuated in MC4R-deficient individuals compared with lean controls in a previous study[22], as this difference emerges 30 min after meal initiation, it is unlikely to contribute to the effects on food preference seen in this study.

The additional experimental flexibility afforded by investigating humans in this study enabled us to dissociate consumption from subjective liking. Interestingly, the increased consumption of high fat food in MC4R deficient individuals did not appear to be related to increased subjective liking. One possible explanation is that the motivating effect of high fat content is implicit and non-conscious, consistent with a theoretical distinction between 'liking' and 'wanting'[23]. However, a failure to observe an effect on rated (subjective) liking may be an issue of power and/or reflect the choice of scaling method used.

What is the potential physiological relevance of these findings? In free living (nutritionally replete) humans, food preference is complicated by the fact that sweet foods are often also high fat

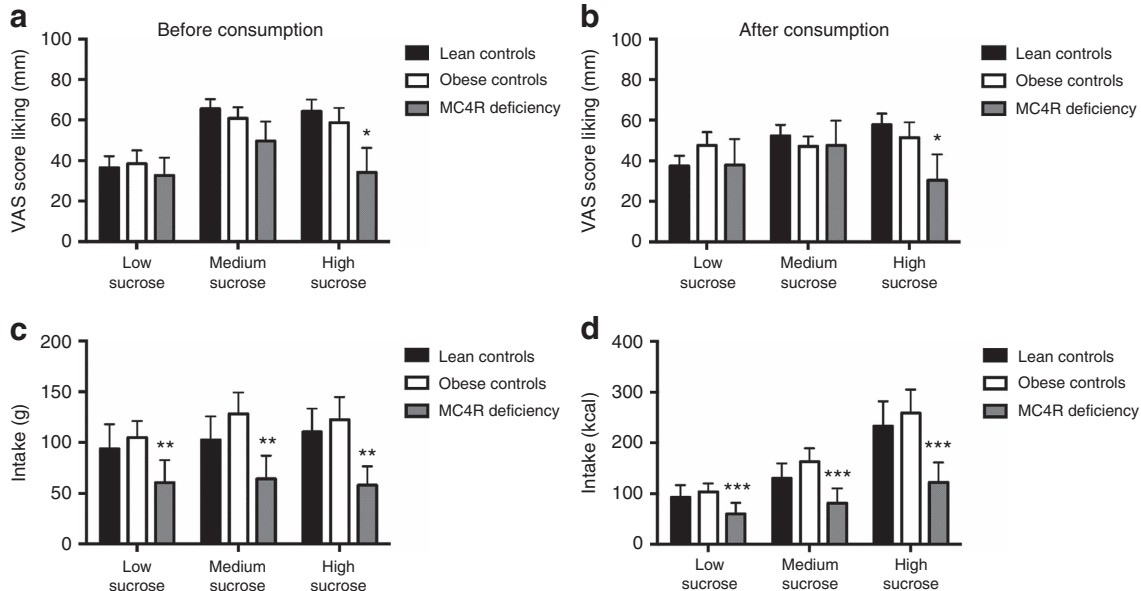

**Figure 2 | Liking ratings and food intake in the sucrose preference test.** Liking of low, medium and high sucrose food before (**a**) and after (**b**) an *ad libitum* meal test in lean controls (*n* = 20), obese controls (*n* = 20) and individuals with MC4R deficiency (*n* = 10). Liking ratings for the high sucrose meal were significantly reduced in MC4R deficiency. Means ± s.e.m. (error bars) are shown. Results were analysed using a linear mixed-effects model. **P = 0.0252. Total intake (g and kcal; **c,d** respectively) in lean controls, obese controls and individuals with MC4R deficiency. **P = 0.0064 and ***P = 0.0031 for group in a linear mixed-effects model.

energy dense foods, thus the choice is rarely between a high fat food and a high sucrose food. We hypothesize that the preference for energy-dense high-fat foods in animals and humans may represent part of the adaptive response to nutritional deprivation. Fasting/starvation leads to a fall in leptin levels which triggers physiological and behavioural responses to restore energy homoeostasis[24]. In keeping with this hypothesis, leptin-deficient *ob/ob* mice provided with single energy-source diets self-select a higher proportion of energy from high-fat (94%) and a lower proportion from high-carbohydrate food (3%) (ref. 25). Critical to mediating the response to fasting/starvation is an increase in the endogenous MC4R antagonist AgRP, which potently stimulates food intake[26], increases fat preference and reduces carbohydrate intake when administered centrally[27]. Thus, the preference for fat (which delivers twice as many calories/gram as carbohydrates or protein and can be readily stored in adipose tissue), at the expense of sugars/carbohydrates may represent an advantageous behaviour that is expressed in the face of nutritional depletion. Further studies of food preference in the weight reduced/partially-leptin deficient state will be needed to test this hypothesis. The detailed behavioural study of humans with naturally occurring mutations affecting specific central signalling pathways provides a powerful tool to link contemporary advances in the understanding of the molecular control of energy balance to the biology underpinning food preference in humans.

## Methods

**Recruitment and ethical committee approval.** All studies were approved by the Cambridge Local Research Ethics Committee (03/103) and undertaken after informed consent. Lean and obese individuals were recruited from the local population by advertisement and MC4R deficient individuals were recruited from our existing cohort (the Genetics of Obesity Study[20]). Exclusion criteria were the use of any medication, any medical or psychiatric disease and food allergies/intolerances.

**Fat preference test.** We developed a three choice, *ad libitum* meal paradigm (Chicken Korma and Rice) in which fat content was covertly manipulated

(by adding rapeseed oil) to provide 20 (low), 40 (medium) and 60% (high) of the total caloric content (Table 1), critically without altering appearance, texture or taste. The test was conducted 4 h after a standardized breakfast which provided 20% of individual energy requirements. Participants were initially provided with 'tasters' (15 g) of the three meals and liking/disliking ratings obtained using VAS before meal consumption. We then provided large dishes containing the low/medium/high fat chicken korma curry and rice (finely chopped and mixed to prevent selection of specific components) in excess (10 MJ; ~2,390 kcal). Participants were encouraged to try the three dishes without explicitly stating that the three meals were different and instructed to eat until they were comfortably full. The amount consumed was covertly weighed. After the meal, liking ratings were measured as before.

**Sucrose preference test.** The test dessert consisted of Eton mess (sweet British pudding consisting of yoghurt, cream, meringues and strawberries) to provide 8% (low) sucrose (Table 3). We then added caster sugar (sucrose) to increase the energy content from sucrose to 26 (medium) and 54 (high). To avoid interference from fat preference in this paradigm, the energy content from fat was clamped at ~30% (Table 3). Participants were admitted to the Clinical Research Facility at 3:00 pm after a standardized breakfast and lunch. Participants were initially provided with 'tasters' (15 g) of the three desserts and liking/disliking ratings obtained using VAS before and after consumption. Participants were invited to try all three desserts, provided in excess, and eat until they were comfortably full. The amount consumed was covertly weighed.

**Sensory discrimination test.** To test whether there was a perceptible sensory difference between the low (20%; A) and high (60%; B) test meals, we employed a Triangle Test using the American Society for Testing and Materials (ASTM) standard protocol[28]. Seventy-eight healthy volunteers received a triad of samples (15 g) coded with a three-digit random number. Each set had two identical samples and one odd sample, presented simultaneously in a predetermined counterbalanced order. Assessors were asked to record which they believed to be the 'odd' sample. If the participant was unable to determine which sample was different, they were asked to guess, since there was a possibility that they were not consciously aware that they could taste the difference. Samples were randomly presented to avoid positional bias since the middle sample is usually chosen as odd. Possible combinations of samples were: AAB, ABA, BAA, BBA, BAB and ABB. Between tasting samples, participants were asked to take a sip of water to avoid sensory fatigue.

**Statistical analysis.** Results were analysed using either ANOVA analysis (with interaction terms for study group and study meal) with Tukey's HSD *post-hoc* comparisons for the fat preference study or a linear mixed-effects model

for the sucrose preference study. Significance was set at $P = 0.05$. All statistical analyses were performed in R.

**Data availability statement.** All relevant data are available from the authors.

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

## Acknowledgements

This work was supported by the Wellcome Trust (to A.A.v.d.K., P.C.F., I.S.F.), the National Institute for Health Research Cambridge Biomedical Research Centre (to S.O'R., I.S.F.), the Bernard Wolfe Health Neuroscience Fund (to A.A.v.d.K., I.S.F., P.C.F.) and the European Research Council (I.S.F.). This work was supported by the NeuroFAST consortium which is funded by the European Union's Seventh Framework Programme (FP7/2007–2013) under grant agreement no 245009.

## Author contributions

I.S.F. and A.A.v.d.K. designed the study; all authors recruited participants, conducted the research and acquired data; A.A.v.d.K., N.S. and I.S.F. analysed data and performed statistical analyses; A.A.v.d.K., P.C.F and I.S.F. wrote the paper; all authors contributed to and approved the paper.

## Additional information

**Competing financial interests:** The authors declare no competing financial interests.

**How to cite this article:** van der Klaauw, A. A. *et al.* Divergent effects of central melanocortin signalling on fat and sucrose preference in humans. *Nat. Commun.* **7**, 13055 doi: 10.1038/ncomms13055 (2016).

