## [Peer Review File · Nature Communications]

Reviewers' comments:

Reviewer #1 (Expert in the neurobiology of food intake; Remarks to the Author):

In this short manuscript, the authors present the results of a study comparing taste preferences and food intake in people who are heterozygous carriers of loss of function mutations in the MC4R gene. Two separate studies were performed. In the first study, the "liking rating" and total intake was assessed for three diets which differed in fat content, but which were otherwise indistinguishable. The meal was provided in excess at mid-day, with the participant provided a standardized breakfast amounting to 20% of daily energy requirements. In the second study, sucrose preference was tested using a dessert with low, medium or high sucrose content.

The main findings were as follows. In the first experiment, the lean and obese controls (LC, OC) consumed equal amounts of the low, medium or high fat meals. However, MC4R haploinsufficiency was associated with a higher intake of the high fat meal (and less of the low- and medium-fat meals). There was no significant difference in VAS score, suggesting a subconscious decision to consume more of the high fat meal. In the second experiment, MC4R haploinsufficiency was associated with reduced VAS score for the high sucrose dessert, and lower intake of all three diets.

The originality and importance of this study stems from its clinical nature. As the authors mention (although their review of the literature is rather selective and not extensive, probably owing to space constraints) evidence of altered preference has been presented in the rodent literature. The approach used appears to be sound, and the data appear to be of high quality. There are some caveats that will be addressed later in the review. As expected given the experience of this group, the statistical analysis used is appropriate.

The conclusions reached by the authors appear to be sound. Given that a similar phenotype has been reported in rodent models, the outcomes certainly appear to be valid. The experience of the group involved in the studies and their unique capacity for performing these experiments speaks to validity and reliability.

In my view, the manuscript could be improved by showing intake as kcal, not grams (or perhaps show both). In Fig. 1C, the authors make the point that "total intake" did not differ between the groups, but don't show the data. They should report the caloric content of the diets in table 1 - it is hard to see how the high fat diet will not have a higher caloric density. If this is indeed the case, then in Fig. 1C total caloric intake will be increased in the MC4R haploinsufficient group. This is a minor point, but suggests it would be consistent with a satiety effect (in other words, the MC4R haploinsufficient group will consume a larger meal based on calories). I can understand why the authors chose to present the data in grams, because it probably accentuates the food selection differences without the potential confound of higher caloric intake.

I also had an issue with the model/hypothesis presented in the last paragraph. I can understand, and probably agree with the hypothesis that low leptin drives a preference towards calorie-dense high fat meals. However, this theory is not consistent with Cone's PNAS paper where the MC4RKO actually more chow and (proportionately) less HFD (Fig. 5D, E in that article). If the AgRP/MC4R interaction was driving mice (and humans) towards increased preference for fat-rich foods, then wouldn't the opposite be true? Also, has anyone reported the results of experiments examining food preferences in weight-normal and weight-loss situations, demonstrating a comparable drive towards consuming high fat meals? Moreover, given that MC4Rs may regulate the secretion of gut "satiety" peptides, measurements of neuroendocrine factors and metabolic outcomes (e.g., NEFA, TG,

glucose) would have been useful. From that perspective, the results presented are preliminary, with further studies comparing preferences for fat-rich foods in normal and weight loss condition. As mentioned above, the authors have elected to cite a few key papers in the field. Other groups have examined preferences in SNP studies, however they are weaker in that they used food diaries that are less reliable than the objective approach used in this paper.

Reviewer #3 (Expert in cognitive psychology and food preferences; Remarks to the Author):

In this paper van der Klaauw and colleagues examine fat and sucrose preference in healthy-weight, and obese individuals with and without heterozygous loss-of-function MC4R variants. Although this is a rare mutation, the topic is of interest because disrupted melanocortin signaling in animals has been associated with increased fat preference and reductions in sucrose consumption.

Consistent with animal work the authors report that individuals with a loss-of-function mutation in MC4R exhibit robust increased intake of a high fat meal and decreased intake of a high sucrose dessert. This implicates the melanocortin system in taste preference in humans; however, the primary novelty of the study is the opportunity to dissociate "liking" and intake. Unfortunately, there are several major problems with the perceptual assessment that undermine the interpretation of this data.

1. They do not measure fat/sugar sensitivity or intensity perception so that it is not possible to rule out decreases or increases in oral sensations as driving effects. For example, intake may be greatest for the highest fat concentration because individuals with impaired melanocortin signaling do not detect the lower concentrations of fat. At the same time meal liking might be associated with other sensory properties of the foods that were not assessed. In this case the interpretation would be that the mutation influences sensitivity rather than preference or liking per se.

2. A second issue is that the authors describe the meal as "identical in appearance and taste" and claim that adding additional fat "did not alter appearance, texture and taste". They cannot make this claim without performing a discrimination test - such as a triangle test. Without this they can only claim that the meals were rated similarly on a number of dimensions. This is problematic for the interpretation that the effect is related to an implicit or unconscious process. A further problem is that perception varies widely between individuals and the participants in the main study are not asked to evaluate the meals on dimensions other than liking.

3. There is not enough information provided to evaluate the meals or the independent validation study. For example, it is not clear what the participants were rating. How does one rate the "look" of a meal with a Likert scale? Where there descriptors on the scale? Do participants actually discriminate taste from smell? Most people have a difficult time with this. Are the participants rating intensity or liking or something else? From the graph it appears that the low fat meal does differ on a number of dimensions. Were these trends? What statistical tests were used?

Given that the novelty of this work rests on the perceptual data these three concerns detract from the overall significance of the work. However, given the potential importance of such a finding I would encourage the authors to consider performing more rigorous testing (e.g. sensitivity,

intensity, discrimination.)

Other concerns/suggestions:

4. It is odd that meal consumption does not change liking ratings for any of the groups. This suggests that the scale is insensitive because liking ratings should decrease following satiety.

5. Was there an effect of group on the influence of meal on liking ratings for sucrose? Looking at the figure it seems that the middle sucrose concentration decreases in liking for the lean and obese but not the MC4R group.

6. I would encourage the authors to consider an alternative interpretation of their data. More specifically, I wonder if the melanocortin disruption reflects alterations in sugar rather than fat sensing. For example, impaired melanocortin signaling may disrupt sensing the post-ingestive effects of sugar, which results in an "enhanced" preference for fat because this is the only "effective" energetic signal. Here the increased intake of fat would be associated with a hedonic contrast rather than an enhancement of fat reward (the rats are insensitive to sugar reward so they consume primarily fat). This interpretation is consistent with the finding that disruption of melanocortin signaling reduces consumption of sucrose but not sucralose and with the possibility that liking for the sweet meal decreases in controls but not MC4R (comment 5).

7. Did BMI differ between obese and MC4R groups? Were individual differences in BMI related to intake/ratings (i.e. correlations present). What happens when BMI is included as a covariate?

Reviewer #4 (Expert in cognitive psychology and food preferences; Remarks to the Author):

NCOMMS-16-02879

Here the authors attempt to translate animal data on the role of Mc4r expressing neurons on food choice to human participants who have function polymorphisms in MC4R that result in a deficient phenotype. These data are highly novel, and interesting, and the conclusions appear generally justified by the data. However, I do have some concerns that are outlined below.

Major Items

- 1) Food choice and food preference. These terms have specific meanings within the ingestive behavior and sensory/consumer science research communities, and are not entirely synonymous. Accordingly, the manuscript should be carefully reviewed for phrasing each time these appear in the manuscript. The authors may also want to specifically define them operationally (although I also recognize the strict length / word limits for this journal.)
- 2) In figure 1C, could the MC4R individuals have consumed more because of habit and/or higher BMI rather than genetics?
- 3) Use of humans here is a major strength, both in a translational sense, but also as it potentially deconfounds liking from making inferences about liking from intake. As such, I don't disagree with the interpretation that intake of high fat food by the MC4R participants may be implicit. However, it

should also be acknowledged that the failure to observe an effect on rated (subjective) liking could also be an issue of power and/or scaling method. In studies that explore differences in food liking, visual analogue scales are typically used with n's closer to 120, versus the much lower numbers here. Also, the choice of scales for measuring food liking and their relative sensitivity remain a highly debated issue. For examples and discussion, see chapter 8 in Lawless 2013. ISBN 978-0-470-67346-1, as well as Lim. 2011. Food Qual Pref, and Bartoshuk, et al 2006. Phil Trans Royal Soc B. Thus, this interpretation really should be softened somewhat.

4) In the modern western diet, most 'sweets' are really high fat energy dense foods where most of the calories come from fat, so the choice in free living humans is rarely between a high fat food and a high carbohydrate food. Accordingly, the penultimate sentence of the paper should be tempered somewhat.

5) Supplemental Figure 1 is functionally useless due to grossly inadequate power. To borrow the phrase popularized by Carl Sagan, 'absence of evidence is not evidence of absence'. Accordingly, this figure should be removed entirely, as it only serves to mislead the reader.

6) In figure 1, switching the coded scheme / axis nesting between panels 1a/1b and 1c is confusing. Is there are reason they aren't all uniform?

Minor

Pg 4 - VAS is typically visual analogue scale, not score. Please revise.

Pg 4 "or a mixed model." is not clear as written. Consider revising.

Pg 4 - pleasant or liked? What were the exact scale labels and instructions? If liking was assessed, the text should say liking/disliking.

Re: **Divergent effects of central melanocortin signalling on fat and sucrose preference in humans – van der Klaauw et al; # NCOMMS-16-02879**

Responses to Reviewers:

Reviewer #1 (Expert in the neurobiology of food intake; Remarks to the Author):

In this short manuscript, the authors present the results of a study comparing taste preferences and food intake in people who are heterozygous carriers of loss of function mutations in the MC4R gene. Two separate studies were performed. In the first study, the "liking rating" and total intake was assessed for three diets which differed in fat content, but which were otherwise indistinguishable. The meal was provided in excess at mid-day, with the participant provided a standardized breakfast amounting to 20% of daily energy requirements. In the second study, sucrose preference was tested using a dessert with low, medium or high sucrose content. The main findings were as follows. In the first experiment, the lean and obese controls (LC, OC) consumed equal amounts of the low, medium or high fat meals. However, MC4R haploinsufficiency was associated with a higher intake of the high fat meal (and less of the low- and medium-fat meals). There was no significant difference in VAS score, suggesting a subconscious decision to consume more of the high fat meal. In the second experiment, MC4R haploinsufficiency was associated with reduced VAS score for the high sucrose dessert, and lower intake of all three diets. The originality and importance of this study stems from its clinical nature. As the authors mention (although their review of the literature is rather selective and not extensive, probably owing to space constraints) evidence of altered preference has been presented in the rodent literature. The approach used appears to be sound, and the data appear to be of high quality. There are some caveats that will be addressed later in the review. As expected given the experience of this group, the statistical analysis used is appropriate.

We thank this Reviewer for their careful consideration of this manuscript. References were limited in line with the size of the manuscript and space constraints. Nonetheless, in preparing this revision we have addressed a number of additional points and in doing so have added references relating to melanocortin signalling in rodents and humans, food preference in humans and the role of the amygdala in macronutrient preference.

The conclusions reached by the authors appear to be sound. Given that a similar phenotype has been reported in rodent models, the outcomes certainly appear to be valid. The experience of the group involved in the studies and their unique capacity for performing these experiments speaks to validity and reliability.

We appreciate this Reviewer's comments regarding the validity of this work.

In my view, the manuscript could be improved by showing intake as kcal, not grams (or perhaps show both).

We agree with this Reviewer and have now included the fat and sucrose intake as both grams and kcal in revised Figures 1 and 2 (added panel D).

In Fig. 1C, the authors make the point that "total intake" did not differ between the groups, but don't show the data.

We thank the reviewer for their comments and have now included the data for total intake in the text as follows (pg 5):

“There was no difference in mean total intake (+/- SEM; g) between lean controls (488.5 +/- 43.1), obese controls (551.0 +/- 48.8) and MC4R deficient individuals (561.5 +/- 60.6); P=0.53 for the group comparison”.

They should report the caloric content of the diets in table 1 - it is hard to see how the high fat diet will not have a higher caloric density. If this is indeed the case, then in Fig. 1C total caloric intake will be increased in the MC4R haploinsufficient group. This is a minor point, but suggests it would be consistent with a satiety effect (in other words, the MC4R haploinsufficient group will consume a larger meal based on calories). I can understand why the authors chose to present the data in grams, because it probably accentuates the food selection differences without the potential confound of higher caloric intake.

We have now added the caloric density to Supplementary Table 1 and 3 and included the fat and sucrose intake as both grams and kcal in revised Figures 1 and 2 (panels C and D) so readers can assess both parameters for the reasons this Reviewer highlights.

I also had an issue with the model/hypothesis presented in the last paragraph. I can understand, and probably agree with the hypothesis that low leptin drives a preference towards calorie-dense high fat meals. However, this theory is not consistent with Cone's PNAS paper where the MC4RKO actually more chow and (proportionately) less HFD (Fig. 5D, E in that article). If the AgRP/MC4R interaction was driving mice (and humans) towards increased preference for fat-rich foods, then wouldn't the opposite be true?

Also, has anyone reported the results of experiments examining food preferences in weight-normal and weight-loss situations, demonstrating a comparable drive towards consuming high fat meals?

The Reviewer makes an interesting and important point. Some of the findings in Roger Cone's paper are discordant with our data and similarly discordant with other work in rodent models of disruption of melanocortin signalling. The explanation for this discrepancy is unclear but may reflect methodological differences. We agree that further studies will be needed to test food preference in a forced choice paradigm in MC4R deficient humans to see whether preference is influenced by the composition of the “alternative choice” meal.

Additionally, we agree that studies that test fat/sucrose preference in the weight reduced/leptin deficient state will be needed to directly test this hypothesis. Whilst we are undertaking such studies in the light of our findings in MC4R deficiency, these studies will take substantially more time to complete and as such are beyond the scope of this manuscript. We hope that publication of our work will similarly stimulate others to test this hypothesis. We have added a comment to this effect.

We have added the following comments to the Discussion:

Pg 6; “Additional studies using a forced choice paradigm will be needed to test whether disruption of melanocortin signalling primarily alters sucrose or fat sensing.”

Pg 7; “Further studies of food preference in the weight reduced/partially-leptin deficient state will be needed to test this hypothesis.”

Moreover, given that MC4Rs may regulate the secretion of gut "satiety" peptides, measurements of neuroendocrine factors and metabolic outcomes (e.g., NEFA, TG, glucose) would have been useful.

We agree that it is important to address whether differences in neuroendocrine, metabolic parameters or pre-meal gut hormone levels might potentially contribute to differences in food preference between the groups studied. Of note, in a number of previous studies we have not found differences in gut hormones (total ghrelin, PYY, GLP-1) nor in neuroendocrine or metabolic factors (NEFA, lipids, insulin, glucose, thyroid hormones) in MC4R deficiency compared to obese controls (Farooqi et al NEJM 2003; Greenfield et al NEJM 2009; van der Klaauw et al JCEM 2013 and unpublished data on 535 patients with loss of function MC4R variants).

Another question is whether impaired central MC4R mediated signalling may alter the meal-related secretion of gut peptides. Indeed, we have previously addressed this question in dynamic meal-related studies in MC4R deficient individuals, lean controls and obese controls. We found that although pre-meal total ghrelin levels were comparable to weight-matched controls, postprandial total ghrelin suppression was attenuated in MC4R-deficient individuals compared to lean controls ($P < 0.05$) (van der Klaauw et al JCEM 2013); a similar pattern was observed in comparison to obese controls, but this difference did not reach statistical significance. As this difference in postprandial ghrelin emerges at 30 minutes, it is unlikely to contribute to the effects on food preference seen in this study (ie: too late to affect consumption of the test meal). We have now discussed this topic and have included the appropriate references as follows:

Pg 5; “Previous studies have shown that pre-prandial gut hormone levels (total ghrelin, PYY, GLP-1) as well as plasma insulin and glucose are comparable in MC4R deficiency and weight-matched controls. Although postprandial total ghrelin suppression was attenuated in MC4R-deficient individuals compared to lean controls in a previous study, as this difference emerges 30 minutes after meal initiation, it is unlikely to contribute to the effects on food preference seen in this study.

From that perspective, the results presented are preliminary, with further studies comparing preferences for fat-rich foods in normal and weight loss condition.

We agree that additional studies in the weight reduced state will be of great interest but we consider that they are beyond the scope of this manuscript. We have added a comment to this effect.

Pg 7; “Further studies of food preference in the weight reduced/partially-leptin deficient state will be needed to test this hypothesis”.

As mentioned above, the authors have elected to cite a few key papers in the field. Other groups have examined preferences in SNP studies, however they are weaker in that they used food diaries that are less reliable than the objective approach used in this paper.

We appreciate that it would be important to cite other papers examining genetic associations with food choice/preference for completeness, although as this Reviewer points out these are often studies of common variants of modest effect size with preference extrapolated from food diaries. A sentence to this effect has been added in the Discussion.

Pg 6; “Although common genetic variants in the fatty acid translocase CD36 have been associated with liking of high-fat foods in African Americans and common obesity-associated variants have been associated with diary reports of food choices in a number of studies (17, 18), this is to our knowledge one of the first experimental studies to show a direct association between macronutrient preference and a specific genetic/molecular mechanism in humans.”

Reviewer #3 (Expert in cognitive psychology and food preferences; Remarks to the Author):

In this paper van der Klaauw and colleagues examine fat and sucrose preference in healthy-weight, and obese individuals with and without heterozygous loss-of-function MC4R variants. Although this is a rare mutation, the topic is of interest because disrupted melanocortin signaling in animals has been associated with increased fat preference and reductions in sucrose consumption. Consistent with animal work the authors report that individuals with a loss-of-function mutation in MC4R exhibit robust increased intake of a high fat meal and decreased intake of a high sucrose dessert. This implicates the melanocortin system in taste preference in humans; however, the primary novelty of the study is the opportunity to dissociate "liking" and intake. Unfortunately, there are several major problems with the perceptual assessment that undermine the interpretation of this data.

1. They do not measure fat/sugar sensitivity or intensity perception so that it is not possible to rule out decreases or increases in oral sensations as driving effects. For example, intake may be greatest for the highest fat concentration because individuals with impaired melanocortin signaling do not detect the lower concentrations of fat. At the same time meal liking might be associated with other sensory properties of the foods that were not assessed. In this case the interpretation would be that the mutation influences sensitivity rather than preference or liking per se.

We thank this Reviewer for their careful consideration of this manuscript. On reflection, we think that we should begin addressing the Reviewer’s comments by clarifying our use of the term “preference” (indeed, this also relates to a key point made by Reviewer 4). Specifically, we are using the term operationally to refer to the choice behaviour without making any implications about whether this reflects an explicit preference or increased liking for a

particular option. That is, if a participant chooses one option more than others, we refer to this as a preference. "Liking" on the other hand, is used to denote the score given by the participant on a liking scale.

The Reviewer's central point, and one that we agree with, is that choosing more of the high fat option (i.e. in our terminology, showing a preference for this option) could emerge from several possible differences. It could be that the person explicitly likes the option more (an idea which we reject given the results of the liking ratings); it could be that the option has particular sensory properties that drive choice (or, as reviewer 3 suggests that MC4R patients lack sensitivity to such properties in the low fat meal) or it could be that there is some implicit or unconscious driver of choice. It is always risky to make assertions that processes are truly implicit and, for this reason, although we raise this as a possibility in the revised manuscript, we do so only tentatively. Moreover, we do not feel that a behavioural study such as this one can distinguish between a preference driven by enhanced sensitivity to particular properties for the high fat condition as opposed to a reduced sensitivity to properties of the low fat condition. We therefore feel that it is important to be clear that our findings demonstrate the following:

- a. There is a relative preference (as defined above) for the high fat meal in MC4R deficient people
- b. The preference is not, evidently, driven by increased liking for this option given the results of the liking ratings on the taster test and after the meal.
- c. The preference could emerge from implicit or explicit processing or both.
- d. The preference is specific to fat as opposed to sugar.

2. A second issue is that the authors describe the meal as "identical in appearance and taste" and claim that adding additional fat "did not alter appearance, texture and taste". They cannot make this claim without performing a discrimination test - such as a triangle test. Without this they can only claim that the meals were rated similarly on a number of dimensions. This is problematic for the interpretation that the effect is related to an implicit or unconscious process. A further problem is that perception varies widely between individuals and the participants in the main study are not asked to evaluate the meals on dimensions other than liking.

Ideally we would have liked to undertake a comprehensive assessment of fat/sugar sensitivity in all three groups. Unfortunately, as MC4R deficiency is a rare condition, it was not feasible to perform a series of independent studies in this group. We did not want to perform sensitivity tests in the same participants that were having the food preference tests as this would make them aware of the nature of the test. We agree this is a limitation and have added a note to this effect alongside discussion of the Triangle Test which we performed to address point 2 (see below). However, this does not detract from the main finding that MC4R deficient people prefer the high fat meal, but rather is one potential explanation for this observation.

At this Reviewer's suggestion, we have now performed a sensory discrimination test (Triangle Test) in normal weight controls to test in an unbiased, blinded and randomised manner whether participants were able to distinguish one meal from the other two. Accordingly, we have removed Supplementary Figure 1.

Ideally, we would have liked to perform this test in the MC4R deficient group. However, the low prevalence of MC4R deficiency precluded such a study. We have therefore added the Triangle Test to the Methods section and the Results as follows:

Pg 5; "In a sensory discrimination test, we found that 41 out of the 78 panelists we tested correctly identified the odd sample in the test. According to international standards for this test, in a study with 78 panelists ($\alpha = 0.001$) at least 40 correct responses are required to reject the null hypothesis (ie: that there is no difference between the meals). Since we observed 41 correct responses, we cannot conclude that the two meal types are the same, although these results are suggestive. We were not able to perform this test in a comparable number of people with MC4R deficiency due to the rarity of this disorder. As such we cannot formally exclude the possibility that attenuated perception of fat may have contributed to the effect that we observed but it is noteworthy that on debriefing participants did not report differences between the foods and were unaware that we had manipulated fat content."

Two co-authors who performed this work have been added.

3. There is not enough information provided to evaluate the meals or the independent validation study. For example, it is not clear what the participants were rating. How does one rate the "look" of a meal with a Likert scale? Where there descriptors on the scale? Do participants actually discriminate taste from smell? Most people have a difficult time with this. Are the participants rating intensity or liking or something else? From the graph it appears that the low fat meal does differ on a number of dimensions. Were these trends? What statistical tests were used?

Given that the novelty of this work rests on the perceptual data these three concerns detract from the overall significance of the work. However, given the potential importance of such a finding I would encourage the authors to consider performing more rigorous testing (e.g. sensitivity, intensity, discrimination.)

As above (responses to 1 and 2).

Other concerns/suggestions:

4. It is odd that meal consumption does not change liking ratings for any of the groups. This suggests that the scale is insensitive because liking ratings should decrease following satiety.

We agree that generally liking ratings decrease following a meal and we have observed this in other studies where we used the same scale (van der Klaauw JCEM 2013) 30 minutes after meal consumption. The fact that we did not observe a fall in liking after meal

consumption in this study probably reflects the timing of the assessment which was performed immediately after completion of the ad libitum meal.

5. Was there an effect of group on the influence of meal on liking ratings for sucrose? Looking at the figure it seems that the middle sucrose concentration decreases in liking for the lean and obese but not the MC4R group.

We have examined this and find no significant group*meal ($p=0.14$) or group*meal*time ($p=0.95$) interaction.

6. I would encourage the authors to consider an alternative interpretation of their data. More specifically, I wonder if the melanocortin disruption reflects alterations in sugar rather than fat sensing. For example, impaired melanocortin signaling may disrupt sensing the post-ingestive effects of sugar, which results in an "enhanced" preference for fat because this is the only "effective" energetic signal. Here the increased intake of fat would be associated with a hedonic contrast rather than an enhancement of fat reward (the rats are insensitive to sugar reward so they consume primarily fat). This interpretation is consistent with the finding that disruption of melanocortin signaling reduces consumption of sucrose but not sucralose and with the possibility that liking for the sweet meal decreases in controls but not MC4R (comment 5).

This is an intriguing idea and a possible alternative explanation for our findings. We have now added a comment to this effect in the Discussion as follows:

Pg 6; "Additional studies using a forced choice paradigm would test whether disruption of melanocortin signalling primarily alters sucrose or fat sensing. "

7. Did BMI differ between obese and MC4R groups? Were individual differences in BMI related to intake/ratings (i.e. correlations present). What happens when BMI is included as a covariate?

BMI did not differ significantly between the obese and the MC4R group ($p=0.07$). Notably, this slight trend was driven by one clear outlier in the MC4R group (BMI 54). When this participant was excluded there was no difference between the groups for BMI. A comment has been added to this effect (Pg 4). There was no correlation between BMI and liking ratings.

Reviewer #4 (Expert in cognitive psychology and food preferences; Remarks to the Author):

Here the authors attempt to translate animal data on the role of Mc4r expressing neurons on food choice to human participants who have function polymorphisms in MC4R that result in a deficient phenotype. These data are highly novel, and interesting, and the conclusions appear generally justified by the data. However, I do have some concerns that are outlined below.

Major Items:

1) Food choice and food preference. These terms have specific meanings within the ingestive behavior and sensory/consumer science research communities, and are not entirely synonymous. Accordingly, the manuscript should be carefully reviewed for phrasing each time these appear in the manuscript. The authors may also want to specifically define them operationally (although I also recognize the strict length / word limits for this journal.)

We agree that this is an important distinction and have checked and that we have used the term food preference throughout.

2) In figure 1C, could the MC4R individuals have consumed more because of habit and/or higher BMI rather than genetics?

It is unlikely that the effects we observed could be explained by BMI as this parameter did not differ significantly between the obese control and MC4R deficient groups; a comment to this effect has been added (Pg 4). We agree that it is important to address whether differences in neuroendocrine, metabolic parameters or pre-meal gut hormone levels might potentially contribute to differences in food preference between the groups studied. Of note, in a number of previous studies we have not found differences in gut hormones (total ghrelin, PYY, GLP-1) nor in neuroendocrine or metabolic factors (NEFA, lipids, insulin, glucose, thyroid hormones) in MC4R deficiency compared to obese controls (Farooqi et al NEJM 2003, Greenfield et al NEJM 2009, van der Klaauw et al JCEM 2013; unpublished data on 535 patients with loss of function MC4R variants). We have added a comment as below:

Pg 5; “Previous studies have shown that pre-prandial gut hormone levels (total ghrelin, PYY, GLP-1) as well as plasma insulin and glucose are comparable in MC4R deficiency and weight-matched controls. Although postprandial total ghrelin suppression was attenuated in MC4R-deficient individuals compared to lean controls in a previous study, as this difference emerges 30 minutes after meal initiation, it is unlikely to contribute to the effects on food preference seen in this study. “

3) Use of humans here is a major strength, both in a translational sense, but also as it potentially de-confounds liking from making inferences about liking from intake. As such, I don't disagree with the interpretation that intake of high fat food by the MC4R participants may be implicit. However, it should also be acknowledged that the failure to observe an effect on rated (subjective) liking could also be an issue of power and/or scaling method. In studies that explore differences in food liking, visual analogue scales are typically used with n's closer to 120, versus the much lower numbers here. Also, the choice of scales for measuring food liking and their relative sensitivity remain a highly debated issue. For examples and discussion, see chapter 8 in Lawless 2013. ISBN 978-0-470-67346-1, as well as Lim. 2011. Food Qual Pref, and Bartoshuk, et al 2006. Phil Trans Royal Soc B. Thus, this interpretation really should be softened somewhat.

We are grateful to the Reviewer for highlighting this point and have added the caveat re power and scaling method and softened the interpretation as follows:

Pg 5; The additional experimental flexibility afforded by investigating humans in this study enabled us to dissociate consumption from subjective liking. Interestingly, the increased consumption of high fat food in MC4R deficient individuals **did not appear to be** related to increased subjective liking. **One possible explanation is that** the motivating effect of high fat content is implicit and non-conscious, consistent with a theoretical distinction between “liking” and “wanting” (15). **However, a failure to observe an effect on rated (subjective) liking may be an issue of power and/or reflect the choice of scaling method used.**

4) In the modern western diet, most 'sweets' are really high fat energy dense foods where most of the calories come from fat, so the choice in free living humans is rarely between a high fat food and a high carbohydrate food. Accordingly, the penultimate sentence of the paper should be tempered somewhat.

We agree that this is an important point to highlight. We have now added a sentence to this effect. We have retained the previous sentence which relates to nutritional depletion but highlighted the distinction by commenting that free-living humans are nutritionally replete.

Pg 6; “What is the potential physiological relevance of these findings? In free living (nutritionally replete) humans, food preference is complicated by the fact that sweet foods are often also high fat energy dense foods, thus the choice is rarely between a high fat food and a high sucrose food.”

5) Supplemental Figure 1 is functionally useless due to grossly inadequate power. To borrow the phrase popularized by Carl Sagan, 'absence of evidence is not evidence of absence'. Accordingly, this figure should be removed entirely, as it only serves to mislead the reader.

We appreciate these comments and have removed this Figure. We recognise the comments from this Reviewer and Reviewer 3 regarding the value of measuring sensory discrimination using a test such as the Triangle Test which we have now performed in normal weight controls (see above).

6) In figure 1, switching the coded scheme / axis nesting between panels 1a/1b and 1c is confusing. Is there are reason they aren't all uniform?

We have now changed the Figures so that the colour coding of groups is uniform.

Minor

Pg 4 - VAS is typically visual analogue scale, not score. Please revise.

We have done so throughout.

Pg 4 "or a mixed model." is not clear as written. Consider revising.

We have clarified this statement re the statistical analysis.

Pg 4 - pleasant or liked? What were the exact scale labels and instructions? If liking was assessed, the text should say liking/disliking.

Pg 4; We have changed the explanation to "liking/disliking" as the instructions were specific to "liking".

Pg 4; "Although liking scores for the high fat meal were comparable to those for the low and medium fat meals, ~~as subjectively more pleasant~~"

Reviewers' comments:

Reviewer #1 (Remarks to the Author):

The authors have responded adequately to my comments, and have also performed an additional experiment in response to comments by reviewer 2. I have no further concerns.

Reviewer #3 (Remarks to the Author):

The authors have been responsive to my concerns. There are two small issues they may want to address. First it would be nice to know what type of fat they are using. Second, on page 6 the revised text beginning "Additional studies..." comes out of left field. I think readers will have a hard time with this unless there is more context. For example, they could mention the between subject design and future work should make direct comparisons between fat and sugar.

Reviewer #4 (Remarks to the Author):

NCOMM Review

In the revised manuscript, the authors have addressed all my major concerns. I do have a few final thoughts for their consideration.

1) If one superimposes the Post consumption data on the Pre consumption data, it looks like the MC4R group shows a different pattern of response relative to the two control groups. Specifically, it looks like the MC4R individuals may like the medium and high fat foods more in the post condition as compared to the pre condition, whereas the two control groups show no change, or drop in liking from the pre to post condition. (And all three groups show a drop in liking for the low fat foods moving from pre to post). In light of this, and given the significant differences in intake seen in panel 1C, real differences in liking may be obscured by noise in the VAS measure. Despite the risk of data snooping, I would like to suggest an alternative analysis strategy. Specifically, at a per person level I would try subtracting liking data in the pre condition from the post condition to create a liking change score, and then run the group by fat level ANOVA on those change scores. It may help resolve the discrepancy between the liking and intake data.

2) The statement "this is to our knowledge one of the first experimental studies to show a direct association between macronutrient preference and a specific genetic/molecular mechanism in humans." would seem to need some qualification, given that alcohol is a macronutrient. Specifically, in nondependent individuals, gene variants have been repeatedly associated with differences in alcohol intake (e.g., PMID: 21163912) via sensory mechanisms (e.g., PMID: 26785164). For a review of these data (as well as data on vegetable intake, which is not a macronutrient per se), the authors may want to read PMID: 23878414.

3) Just in case the authors plan to test these participants further, it is actually possible to make stimuli that differs dramatically in fat content while holding texture constant (e.g., PMID: 18353432).

Minor:

Pg 4, new text, last sentence on sensitivity and proportion of distinguishers: The authors are to be commended for adding these data (and for conducting a triangle test correctly). However I would suggest deleting this last sentence, as proportion of distinguishers is jargon among applied sensory practitioners that isn't relevant to the broad readership of this journal. In short, it is needless detail that doesn't add to the paper.

Pg 5, top. The new added bold text about mean intake is confusing as written. Does the comparison refer to intake collapsed across fat level? If so, this should be explicitly stated.

Pg 5, topic sentence, last paragraph: If the authors can spare a citation, it would be good to provide one here to support this statement. While true, it is still surprisingly contentious to some.

Pg 6, "primarily alters sucrose or fat sensing... OR the reward from these stimuli." That is, differences in intake could be driven by differences in sensation or affective response to those sensations. You cannot tell from the present data.

Re: **Divergent effects of central melanocortin signalling on fat and sucrose preference in humans – van der Klaauw et al; # NCOMMS-16-02879A**

Responses to Reviewers:

Reviewer #1 (Remarks to the Author):

The authors have responded adequately to my comments, and have also performed an additional experiment in response to comments by reviewer 2. I have no further concerns.

We are pleased that the Reviewer considers that we have addressed all their comments.

Reviewer #3 (Remarks to the Author):

The authors have been responsive to my concerns. There are two small issues they may want to address. First it would be nice to know what type of fat they are using.

Now added to the Methods (rapeseed oil).

Second, on page 6 the revised text beginning "Additional studies..." comes out of left field. I think readers will have a hard time with this unless there is more context. For example, they could mention the between subject design and future work should make direct comparisons between fat and sugar.

We are pleased that the Reviewer considers that we have addressed their comments. We have modified this statement to clarify the point and have moved some of the key points in the Discussion to improve the flow of the manuscript (in view of additional subheadings requested by the Journal).

Reviewer #4 (Remarks to the Author):

NCOMM Review

In the revised manuscript, the authors have addressed all my major concerns. I do have a few final thoughts for their consideration.

We are pleased that the Reviewer considers that we have addressed their comments.

1) If one superimposes the Post consumption data on the Pre consumption data, it looks like the MC4R group shows a different pattern of response relative to the two control groups. Specifically, it looks like the MC4R individuals may like the medium and high fat foods more in the post condition as compared to the pre condition, whereas the two control groups show no change, or drop in liking from the pre to post condition. (And all three groups show a drop in liking for the low fat foods moving from pre to post). In light of this, and given the significant differences in intake seen in panel 1C, real differences in liking may be obscured by noise in the VAS measure. Despite the risk of data snooping, I would like to suggest an alternative analysis strategy. Specifically, at a per person level I would try subtracting liking data in the pre condition from the post condition to create a liking change score, and then run the group by fat level ANOVA on those change scores. It may help resolve the discrepancy between the liking and intake data.

This is an interesting idea. As suggested, we derived a liking change score and then ran the group by fat level analysis (ANOVA). As shown in the Figure included for the Reviewer, this does not indicate a significant effect of fat level (meal) * group on change score as a similar pattern is seen in MC4R deficient individuals and obese controls.

2) The statement "this is to our knowledge one of the first experimental studies to show a direct association between macronutrient preference and a specific genetic/molecular mechanism in humans." would seem to need some qualification, given that alcohol is a macronutrient. Specifically, in nondependent individuals, gene variants have been repeatedly associated with differences in alcohol intake (e.g., PMID: 21163912) via sensory mechanisms (e.g., PMID: 26785164). For a review of these data (as well as data on vegetable intake, which is not a macronutrient per se), the authors may want to read PMID: 23878414.

We appreciate this information.

3) Just in case the authors plan to test these participants further, it is actually possible to make stimuli that differs dramatically in fat content while holding texture constant (e.g., PMID: 18353432).

We appreciate this information.

Minor:

Pg 4, new text, last sentence on sensitivity and proportion of distinguishers: The authors are to be commended for adding these data (and for conducting a triangle test correctly). However I would suggest deleting this last sentence, as proportion of distinguishers is jargon among applied sensory practitioners that isn't relevant to the broad readership of this journal. In short, it is needless detail that doesn't add to the paper.

We have removed this statement.

Pg 5, top. The new added bold text about mean intake is confusing as written. Does the comparison refer to intake collapsed across fat level? If so, this should be explicitly stated.

We have changed this to “total intake collapsed across fat level (mean +/- SEM; g).”

Pg 5, topic sentence, last paragraph: If the authors can spare a citation, it would be good to provide one here to support this statement. While true, it is still surprisingly contentious to some.

We have added a key reference by Dr Drewnowski here.

Pg 6, "primarily alters sucrose or fat sensing... OR the reward from these stimuli." That is, differences in intake could be driven by differences in sensation or affective response to those sensations. You cannot tell from the present data.

We have modified this sentence as suggested.